# Factors associated with changes in the objectively measured physical activity among Japanese adults: A longitudinal and dynamic panel data analysis

Daiki Watanabe[1,2,3], Haruka Murakami[4], Yuko Gando[2,5], Ryoko Kawakami[1], Kumpei Tanisawa[1], Harumi Ohno[2], Kana Konishi[2,6], Azusa Sasaki[7], Akie Morishita[8], Nobuyuki Miyatake[9], Motohiko Miyachi[1,2]*

1 Faculty of Sport Sciences, Waseda University, Tokorozawa, Saitama, Japan, 2 National Institute of Health and Nutrition, National Institutes of Biomedical Innovation, Health and Nutrition, Shinjuku-ku, Tokyo, Japan, 3 Institute for Active Health, Kyoto University of Advanced Science, Sogabe-cho, Kameoka-city, Kyoto, Japan, 4 Faculty of Sport and Health Science, Ritsumeikan University, Kusatsu-city, Shiga, Japan, 5 Faculty of Sport Science, Surugadai University, Saitama, Japan, 6 Faculty of Food and Nutritional Sciences, Toyo University, Ora-gun, Gunma, Japan, 7 Department of Food and Nutrition, Jumonji University, Niiza, Saitama, Japan, 8 Okayama Southern Institute of Health, Okayama Health Foundation, Okayama-city, Okayama, Japan, 9 Faculty of Medicine, Kagawa University, Kita-gun, Kagawa, Japan

* miyachim@waseda.jp, cardiovascular0327@mac.com

**Data Availability Statement:** Data used in the present study cannot be shared publicly because of ethical regulation and Japanese Personal

## Abstract

Factors associated with dynamic changes in the objectively measured physical activity have not been well understood. We aimed to 1) evaluate the longitudinal change in the physical activity trajectory according to sex which is associated with age and to 2) determine the factors associated with the dynamic change in physical activity-related variables across a wide age range among Japanese adults. This longitudinal prospective study included 689 Japanese adults (3914 measurements) aged 26–85 years, whose physical activity data in at least two surveys were available. Physical activity-related variables, such as intensity (inactive, light [LPA; 1.5 to 2.9 metabolic equivalents (METs)], moderate-to-vigorous [MVPA; $\geq$3.0 METs]), total energy expenditure (TEE), physical activity level (PAL), and step count, were evaluated using a validated triaxial accelerometer. Statistical analysis involved the latent growth curve models and random-effect panel data multivariate regression analysis. During a mean follow-up period of 6.8 years, physical activity was assessed an average of 5.1 times in men and 5.9 times in women. The profiles for the inactive time, LPA (only men), MVPA, step count, PAL, and TEE showed clear curvature, indicating an accelerated rate of change around the age of 70. In contrast, other variables exhibited minimal or no curvature over the age span. The MVPA trajectory was positively associated with alcohol consumption, hand grips, leg power, and trunk flexibility and negatively associated with age, local area, body mass index (BMI), comorbidity score, and heart rate over time. Our results indicated that the physical activity trajectory revealed clear curvature, accelerated rate of change around the age of 70, and determined physical health and fitness and BMI as dynamic factors associated with physical activity changes. These findings may be useful to help support populations to achieve and maintain the recommended level of physical activity.

Information Protection Law. The minimal dataset is available from the NOBIOHN Ethics Committee (contact via irb-office@nibiohn.go.jp) for researchers who meet the criteria for access to confidential data.

**Funding:** This study was supported by a Grant-in-Aid for Scientific Research (C) (JP16K00944; to M. Miyachi) for the purpose of designing the study, and for data collection and analysis.

**Competing interests:** The authors declare that they have no competing interests.

## Introduction

Insufficient physical activity is a leading and modifiable risk factor for a short lifespan worldwide [1–3] and has been increasing in high-income countries over time [4]. Daily step count as an objective index of physical activity for 717,527 people in 111 countries revealed heterogeneity across regions and countries [5]. The difference in physical activity between countries may explain the significant regional differences in the economic burden globally [6].

In 2020, the World Health Organization (WHO) released guidelines on physical activity and sedentary behavior to promote health [7]. Although these guidelines highlighted the importance of regular moderate-to-vigorous-intensity physical activity (MVPA) and reducing sedentary behaviors in all adults [7], global estimates show that one in four (27.5%) adults does not meet the recommendations for physical activity [4]. It is important to evaluate how physical activity changes in individuals over time because a previous study has shown decreased mean trajectory of daily step count in a short span of 2 years in older adults [8]. Some prospective cohort studies have shown that the trajectory of self-reported physical activity data is associated with a risk of mortality [9] and disability [10] in middle-aged and older adults. Therefore, there is a need to evaluate the factors associated with physical activity trajectory to increase or maintain physical activity in adults.

Many longitudinal studies have presented baseline predictors associated with physical activity trajectory [11–13]. However, these studies cannot determine an individual's specific physical activity trajectory with the associated factors of dynamic changes. In addition, although previous longitudinal studies have verified the association between self-reported physical activity trajectory and limited time-varying variables [14–16], the factors of dynamic changes associated with objectively measured physical activity over time in adults have not been well-demonstrated. These findings are essential for increasing and maintaining physical activity with the increasing spread of physical inactivity and sedentary behavior worldwide [4]. In this study, we aimed to 1) evaluate the longitudinal changes in the physical activity trajectory associated with age according to sex and 2) determine the factors of dynamic changes associated with the physical activity-related variables trajectory across a wide age range among Japanese adults. We hypothesized that physical activity may decrease with increasing age. Moreover, we can also determine the factors associated with increased or decreased physical activity over time.

## Materials and methods

### Study population

This longitudinal prospective study utilized data from the cohort study, which has been described previously [17–20]. This cohort study was managed by the National Institute of Health and Nutrition (NIHN) since 2007 and aimed to evaluate the association between lifestyle-related diseases and modifiable risk factors, such as dietary intake and physical activity. This cohort study included 760 adults, aged 26–85 years, who lived in the Tokyo metropolitan area ($n$ = 504) and Okayama prefecture ($n$ = 256) in Japan from 2007 to 2018 (population density: 6168.7 people/km$^2$ in Tokyo and 270.1 people/km$^2$ in Okayama) who agreed to participate in this study. These participants were recruited when they participated in a specific health examination conducted at the Okayama Southern Institute of Health and the NIHN. All participants were requested to participate in a health checkup conducted through an annual face-to-face meeting and a mail survey. The investigations were conducted annually using the same survey content and methodology, and the participants were followed up for a maximum of 12 years (S1 Table in S1 File).

Among the participants who were initially included at baseline (*n* = 760), we excluded individuals with missing data on age and sex (*n* = 1), those whose physical activity could not be measured using an accelerometer (*n* = 5), and those who completed the assessment of physical activity only once (*n* = 65). The final dataset included 689 participants (3914 measurements) whose physical activity information from at least two accelerometer surveys was available. Of all participants in this study, 175 individuals (25.4%) under the age of 65 years with a lower amount of MVPA (less than 3.3 metabolic equivalents [METs]-h/day) received an intervention composed of five-time brief counseling sessions to increase MVPA.

## Evaluation of physical activity

Objective physical activity was measured using a validated triaxial accelerometer (EW4800, Panasonic Co., Ltd, Osaka, Japan) against total energy expenditure (TEE), which measured movement using both the doubly labeled water (DLW) and metabolic chamber methods [21]. Research staff were educated on how to use and handle the accelerometer using the manual. All participants were instructed to wear an accelerometer around the waist upon waking up till bedtime except when swimming, sleeping, and bathing. The participants were asked to wear a triaxial accelerometer at least more than 10 hours per day for 28 days. The days were considered valid when participants wore the wearable devices for more than 10 hours/day from self-reported wearing time based on activity records by the participant himself/herself [22]. However, the objective wear time obtained by accelerometer does not include acceleration data with an intensity of less than 1.1 METs when a participant is completely still (no signal time). Therefore, the calculated wearing time cannot distinguish between no signal time and non-wearing time. The objective wearing time was defined as 24 hours minus non-wearing and no signal time. In addition, we excluded data with wearing time less than 6 hours per day. Those who exceeded this criterion confirmed self-reported accelerometer wearing time exceeding 10 hours in 24 individuals randomly selected from the study population. Therefore, we ultimately determined a valid days of accelerometer data use from two criteria: self-reported wearing time from activity records, and objective wearing time from the accelerometer. Median valid days of accelerometer data included more than 2 weeks in all in-person testing from 2007 to 2018 (S1 Table in S1 File). To calculate the mean physical activity time, the sum of all physical activities surveyed over at last 7 days (including weekdays and weekends) was divided by the number of adhered days. No participants were excluded by these criteria for accelerometer data because those who had the valid days of accelerometer data for less than 7 days were asked to wear the accelerometer again.

The intensity for every minute, basal metabolic rate, step count, and physical activity level (PAL) were determined using the maker's algorithm. We obtained the daily physical activity times corresponding to <1.5 METs (sedentary), 1.5 to 2.9 METs (light intensity physical activity: LPA), and ≥3.0 METs (MVPA). The inactive times were calculated using the sum of sedentary (<1.5 METs) and non-wearing periods, which was calculated as 1440 minutes − wearing periods (daily time spent in sedentary, light, moderate, and vigorous physical activity times) [20]. We included inactive time, LPA, MVPA, TEE, PAL, and step count as objective physical activity-related variables.

## Self-reported covariate

Health information, such as medical history and smoking status, was obtained using self-reported structured questionnaires. Dietary intake was evaluated using the Brief-type self-administered Diet History Questionnaire (BDHQ), which consists of 58 food and beverage items that were validated against dietary records [23]. The diet quality was assessed using a

previously validated Nutrition Rich Food (NRF) index 9.3 score [17, 24]. The research staff checked all questionnaires and interviewed respondents with unanswered questions, unclear responses, or to confirm answers. This NRF 9.3 score ranges from 0 (worst diet quality) to 900 (best diet quality) [17, 24]. Alcohol intake was estimated from the consumption frequency and portion size of each alcohol beverage using a program developed based on the Standard Tables of Food Composition in Japan. Alcohol intake (%) was calculated by dividing energy intake from alcohol by total energy intake and multiplying by 100. Based on the data obtained regarding the comorbidity status of each individual, the comorbidity score was added to obtain a total score (including hypertension, dyslipidemia, diabetes, ischemic heart disease, other heart diseases, cerebrovascular diseases, renal failure, cancer, osteoporosis, and depression) ranging from 0 (no comorbidity) to 10 (poor status).

## Measured covariate

Each participant's body weight was measured while in light clothing (BC-600, TANITA Corp., Tokyo, Japan). The body mass index (BMI) was calculated by dividing the measured body weight (kg) by the square of the height ($m^2$). The waist/hip ratio was calculated as the circumferences of the waist (at the level of the navel) to hip (the greatest posterior protuberance, perpendicular to the long axis of the trunk). Trunk flexibility was measured using a sit-and-reach digital instrument (T.K.K.5112; Takei Scientific Instruments Co., Ltd, Japan). Resting heart rate (HR) was measured using an electrocardiogram mounted on a pulse wave examination device (form PWV/ABI BP-203RPEII, Omron Colin, Kyoto, Japan). Hemoglobin was measured using a colorimetric method that utilizes sodium lauryl sulfate from fasting blood samples (≥12 hours). Leg press power was measured using a leg muscle strength measuring device (Anaeropress 3500, COMBI, Tokyo, Japan). This device measured the unidirectional power production of the leg extensors. Grip strength was evaluated using a Smedley Hand Dynamometer (Grip-D TKK5101, Takei Scientific Instruments, Niigata, Japan). Measurements were taken twice from each hand, and the mean of the highest value of each hand was used.

## Statistical analysis

The participants' characteristics were expressed as numbers and percentages for categorical variables and means and standard deviations for continuous variables. We performed imputation to missing values of covariates from five data sets, which were created using the multiple imputation method that utilizes multivariate imputation through a chained equation (MICE) by R software [25, 26]. The details are shown in S1 Table in S1 File. These missing values were assumed as missing at random.

To identify the longitudinal trajectory from repeat measures of physical activity, we estimated a single mean physical activity trajectory across the group using a sex-stratified model that uses the latent growth curve models (LGCM). In addition, the latent class growth models (LCGM) were applied to assess whether study participants could be classified into multiple trajectory groups through the maximum likelihood method. These analyses were used by the STATA macro TRAJ [27], and to construct trajectory shape with a cubic specification. The best-fitting model for LCGM was identified by estimating models with two to eight latent clusters and comparing them using the sample size of the clusters (≥5%) and the Bayesian Information Criterion as the primary fit index [28].

To calculate the correlation coefficients by repeated measurement and cross-sectional analysis between chronological age and physical activity-related variables, we performed Repeated Measures Correlation by the R software [29] and Pearson's correlation analysis, respectively. We evaluated the accuracy and precision of the mean of the group's physical activity-related

variables trajectory using previously reported equations [30]. These equations were used to estimate the required sample size and periods from within-person variance, between-person variance, and the ratio of within-person to between-person variance.

To evaluate the factors associated with the physical activity trajectory, we used the multivariate regression analysis of random-effect panel data [17], which is a method that evaluates related factors from the longitudinal changes of the dependent variable and the explanatory variable to adjust the between-individual characteristics [31]. To evaluate factors associated with the physical activity trajectory, the multivariate analysis included the age (continuous), sex (female or male), region (urban (Tokyo) or local (Okayama)), BMI (continuous), waist/hip ratio (continuous), comorbidity score (continuous), smoking status (never smoker or past and current smoker), alcohol intake (continuous), energy intake (continuous), NRF 9.3 score (continuous), hemoglobin (continuous), HR (continuous), hand grips (continuous), leg power (continuous), and trunk flexibility (continuous). These variables were selected in reference to covariates used in previous studies [11–13, 32–34]. The variance inflation factor (VIF) was used to avoid multicollinearity in the multivariate regression model and all covariates were maintained VIF ≤5 (S2 Table in S1 File). The results of these analyses were shown in regression coefficients (RC) and 95% confidence interval (CI) for each variable per unit increment. To conduct the sensitivity analysis for results, we conducted a similar analysis using the complete cases dataset.

A two-sided $p$-value <0.05 was considered significant. All analyses were performed using STATA MP version 15.0 (StataCorp LP, College Station, TX, USA) and/or R software 3.4.3 (R Core Team, Vienna, Austria).

## Ethical considerations

This study was conducted according to the guidelines laid down in the 1964 Declaration of Helsinki and all procedures involving research study participants were approved by the Research Ethics Committee of the National Institute of Health and Nutrition (approval no. kenei102-01). Written informed consent was obtained from all participants before data acquisition.

## Results

Table 1 shows the characteristics of participants in the study cohorts. On average, the levels of physical activity-related variables were 1026 minutes/day for inactive time, 354 minutes/day for LPA, 60 minutes/day for MVPA, 10216 steps/day for step counts, 1.58 for PAL, and 1930 kcal/day for TEE. The study participants were older than the excluded population, but a serious bias was not observed (S3 Table in S1 File).

During a mean follow-up period of 6.8 years, physical activity-related variables were assessed 5.1 times ($n = 211$; 1089 measurements) in men and 5.9 times ($n = 478$; 2825 measurements) in women on average. The means of physical activity-related variables by age in both men and women in the population level are shown in Fig 1. With increasing age, the average LPA (only men), MVPA, step count, PAL, and TEE declined, while the inactive times increased. These profiles showed clear curvature, indicating an accelerated rate of change around the age of 70, whereas the LPA (only women) exhibited minimal or no curvature over the age span. Moreover, some separate trajectory groups in the physical activity-related variables were identified (S1, S2 Figs in S1 File). The MVPA trajectory were classified into two groups (S1C and S2C Figs in S1 File; 1: low MVPA trajectory group and 2: high MVPA trajectory group). Although this cohort study had a brief counseling intervention for physical activity in some populations, the proportion of individuals in the high MVPA trajectory group in

**Table 1. Baseline characteristics of demographic and physical activity by sex.**

| | Total (*n* = 689) | | Men (*n* = 211) | | Women (*n* = 478) | |
|---|---|---|---|---|---|---|
| Age [years] [a] | 52.0 | (11.6) | 49.0 | (12.0) | 53.4 | (11.1) |
| Local area [*n* (%)] [b] | 227 | (32.9) | 98 | (46.2) | 129 | (27.0) |
| Body mass index [kg/m$^2$] [a] | 22.5 | (2.9) | 23.4 | (2.7) | 22.1 | (2.9) |
| Waist/Hip ratio [a] | 0.88 | (0.07) | 0.89 | (0.06) | 0.87 | (0.07) |
| No comorbidity [*n* (%)] [b] | 524 | (76.1) | 159 | (75.0) | 365 | (76.5) |
| Smoker [*n* (%)] [b] | 196 | (28.4) | 125 | (59.0) | 71 | (14.9) |
| Alcohol drinker [*n* (%)] [b] | 501 | (72.7) | 181 | (85.4) | 320 | (67.1) |
| Energy intake [kcal/day] [a] | 1819 | (492) | 2094 | (523) | 1697 | (424) |
| NRF 9.3 score [a] | 765 | (69) | 742 | (71) | 775 | (65) |
| Hemoglobin [g/dl] [a] | 13.6 | (1.4) | 15.0 | (1.0) | 13.0 | (1.1) |
| Heart rate [bpm] [a] | 63 | (12) | 64 | (13) | 62 | (11) |
| Hand grips [kg] [a] | 33.5 | (9.3) | 43.2 | (6.1) | 29.2 | (6.8) |
| Leg power [w] [a] | 1081 | (410) | 1488 | (386) | 900 | (266) |
| Trunk flexibility [cm] [a] | 39.1 | (9.8) | 35.9 | (10.6) | 40.5 | (9.1) |
| BMR [kcal/day] [a] | 1225 | (144) | 1346 | (161) | 1171 | (95) |
| TEE [kcal/day] [a] | 1930 | (246) | 2040 | (305) | 1882 | (195) |
| Physical activity level [a] | 1.58 | (0.14) | 1.52 | (0.14) | 1.61 | (0.13) |
| Step counts [steps/day] [a] | 10216 | (3556) | 10528 | (3773) | 10078 | (3450) |
| Inactive time [min/day] [a] | 1026 | (101) | 1089 | (93) | 998 | (92) |
| Sedentary time [min/day] [a] | 220 | (47) | 222 | (51) | 219 | (45) |
| LPA [min/day] [a] | 354 | (94) | 294 | (85) | 381 | (86) |
| MVPA [min/day] [a] | 60 | (27) | 57 | (29) | 61 | (26) |

BMR, basal metabolic rate; LPA, low intensity physical activity; MVPA, moderate-to-vigorous physical activity; NRF, nutrient-rich food; TEE, total energy expenditure

[a] Continuous variables were expressed as mean with (standard deviation).

[b] Categorical variables were expressed as number with (percentage).

the intervention group was similar to that in the non-intervention group (5.1% vs 4.9%). These results suggest that brief counseling intervention did not affect the MVPA trajectory.

We estimated correlation coefficients by repeated measurement and cross-sectional analysis between chronological age and physical activity-related variables (Fig 2). The cross-sectional analysis showed stronger correlations of TEE with chronological age compared with the repeated measurement analysis. For other variables, correlation coefficients calculated by repeated measurement and cross-sectional analysis were similar. S4 Table in S1 File shows the accuracy and precision of physical activity-related variables. Group sizes required for estimating a group's "true" mean physical activity-related variables trajectory within a 95% CI with 2.5% deviation ranged from 63 people (PAL) to 1826 people (MVPA). Four time-points of accelerometer data were required to obtain a correlation coefficient (*r*) of 0.95 between an individual's assessed value and their "true" unevaluated usual mean physical activity-related variables.

We evaluated the factors associated with changes in physical activity using multivariate longitudinal analysis (Table 2). The MVPA trajectory was positively associated with alcohol consumption, hand grips, leg power, and trunk flexibility, while negatively associated with age, local area, BMI, comorbidity score, and HR. Both inactive and sedentary times trajectory are associated with higher BMI (Table 2 and S5 Table in S1 File). Associations of TEE, PAL, and step count change with covariates are shown in Table 3. We demonstrated that the TEE was positively associated with BMI, alcohol consumption, energy intake, hand grips, leg power,

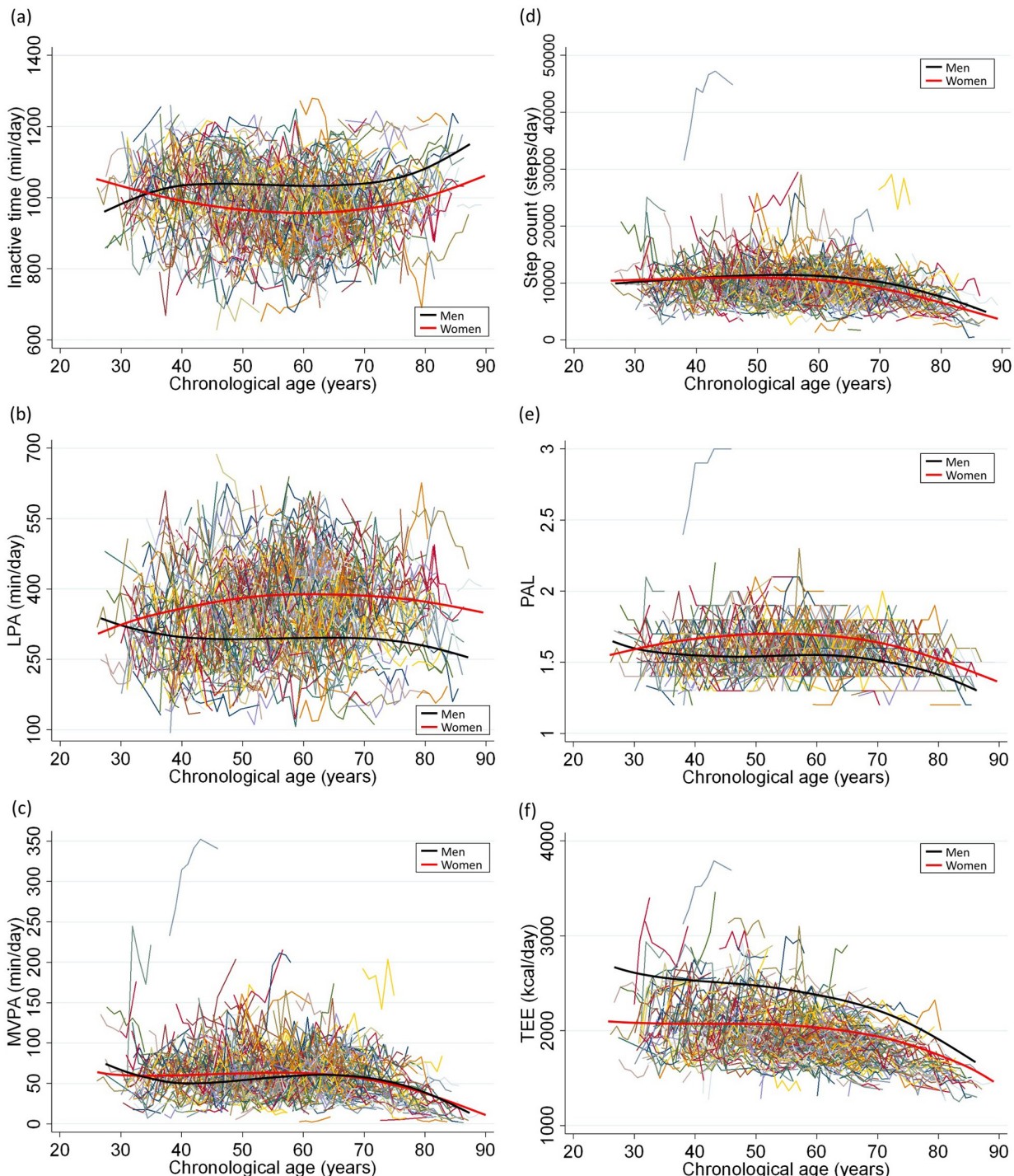

**Fig 1. Longitudinal trajectories of physical activity and total energy expenditure in 689 individuals (3914 measurements).** The latent growth curve models were performed to estimate a single mean (a) inactive time, (b) low intensity physical activity (LPA), (c) moderate-to-vigorous physical activity (MVPA), (d) step count, (e) physical activity level (PAL), and (f) total energy expenditure (TEE) trajectory across the group by a sex-stratified model. Physical activity measurements are presented such that an individual is connected to the same color lines when more than one measurement is assessed for a given individual. Average changes in physical activity with age in this study population were indicated by smooth lines (black for men and pink for women).

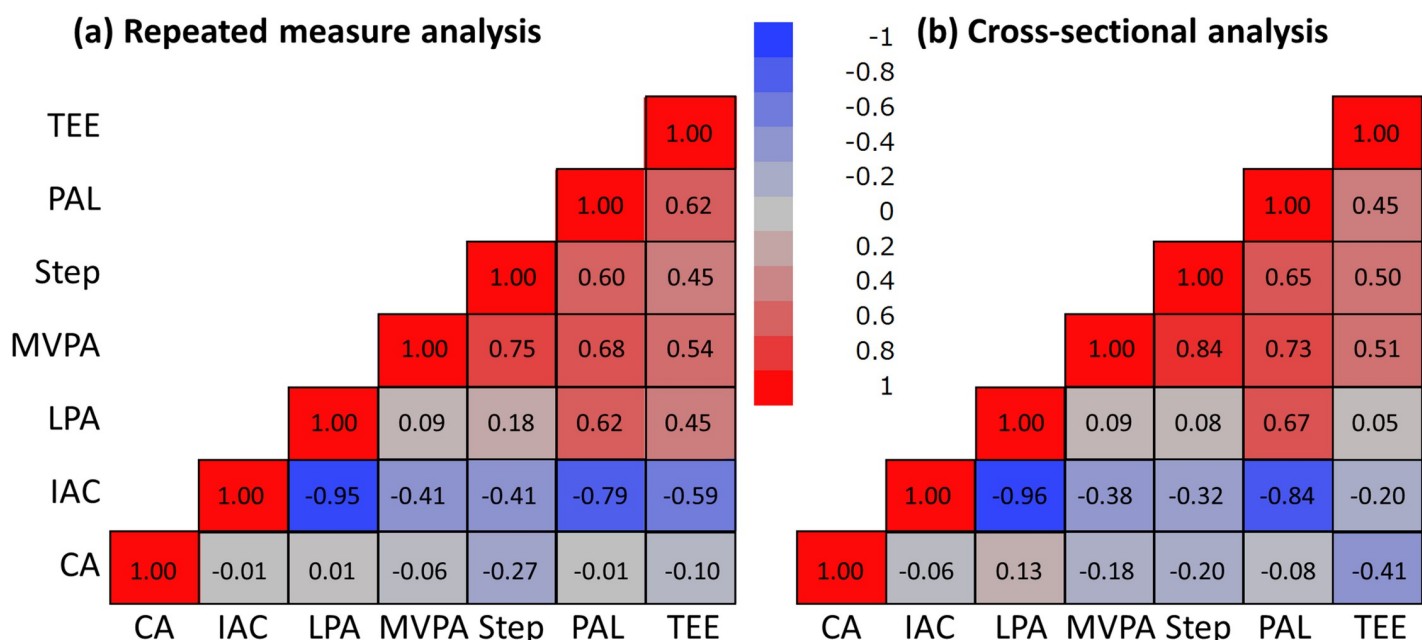

**Fig 2. Comparison of correlation coefficients of physical activity-related variables and chronological age between repeated measurement (a) and cross-sectional data** (b) in 689 individuals (3914 measurements). The red and blue panels were expressed as positive and negative correlation coefficients, respectively. CA, chronological age; IAC, inactive time; LPA, low intensity physical activity; MVPA, moderate-to-vigorous physical activity; PAL, physical activity level; TEE, total energy expenditure.

and trunk flexibility, while negatively associated with age, women, comorbidity score, and HR. These associations such as inactive times, LPA, TEE, PAL, and step counts with covariates were confirmed sex differences. In addition, similar results were obtained for the complete cases, after excluding data with missing values, in the sensitivity analysis (S6 and S7 Tables in S1 File).

## Discussion

In the present study, we indicated that MVPA, step count, PAL, and TEE trajectory revealed clear curvature and accelerated rate of change around the age of 70. Moreover, factors that should be encouraged or limited to improve the physical activity trajectory in Japanese adults were identified. To the best of our knowledge, this is the first study to verify the effect of physical health, fitness, and BMI on the objectively measured physical activity trajectory. These findings may potentially be useful to help increase and maintain physical activity in adults, including the elderly.

Our results demonstrate that MVPA, step count, PAL, and TEE trajectory showed clear curvature and accelerated rate of change around the age of 70 in both sexes. Some longitudinal studies have indicated that increasing age is not associated with decreasing self-reported physical activity trajectory [11, 13] but with decreasing objectively measured physical activity for 2 years in 70 to 90 years-old men [8]. Self-reported physical activity can lead to an increase in systematic reporting bias because individuals may modify their reports in their desired direction without any actual behavior change [35]. Previous studies have reported that longitudinal growth of functional biological age accelerates at around age 70 [36], which supports our results. In addition, all individuals should not be assumed to have the same trajectory because we identified two to five different trajectory groups of physical activity-related variables. Our findings suggest that it is important to provide individuals with opportunities to be able to

**Table 2. Factors associated with changes of physical activity times by multivariate longitudinal analysis.**

| Increment effects/unit | Inactive times | | LPA | | MVPA | |
|---|---|---|---|---|---|---|
| | RC | 95% CI | RC | 95% CI | RC | 95% CI |
| **Total** | ($n$ = 689 [3914 measurement]) | | | | | |
| Within R$^2$ | $R^2$ = 0.013 | | $R^2$ = 0.009 | | $R^2$ = 0.019 | |
| Age [1 year] | -0.583 | (-1.038 to -0.128)* | 0.870 | (0.453 to 1.286)* | -0.288 | (-0.429 to -0.145)* |
| Women sex | -89.084 | (-104.262 to -73.906)* | 91.770 | (77.873 to 105.665)* | -2.639 | (-7.247 to 1.968) |
| Local area | -15.148 | (-29.498 to -0.796)* | 24.339 | (11.200 to 37.476)* | -9.282 | (-13.615 to -4.947)* |
| BMI [1 kg/m$^2$] | 4.143 | (2.502 to 5.783)* | -3.303 | (-4.805 to -1.801)* | -0.804 | (-1.319 to -0.287)* |
| Waist/Hip ratio [1 point] | 42.018 | (-1.372 to 85.408) | -36.946 | (-76.708 to 2.816) | -5.480 | (-19.648 to 8.689) |
| Comorbidity score [1 point] | 3.644 | (0.398 to 6.890)* | -1.521 | (-4.495 to 1.453) | -2.212 | (-3.268 to -1.155)* |
| Smoker | 4.515 | (-4.288 to 13.319) | -2.458 | (-10.523 to 5.607) | -2.053 | (-4.866 to 0.760) |
| Alcohol intake [1% energy] | -0.425 | (-0.931 to 0.081) | 0.159 | (-0.305 to 0.623) | 0.270 | (0.105 to 0.433)* |
| Energy intake [1 kcal/day] | -0.005 | (-0.010 to 0.000) | 0.004 | (-0.001 to 0.008) | 0.001 | (-0.001 to 0.002) |
| NRF 9.3 score [1 point] | 0.002 | (-0.035 to 0.039) | -0.009 | (-0.043 to 0.024) | 0.008 | (-0.004 to 0.019) |
| Hemoglobin [1 g/dl] | -0.423 | (-1.410 to 0.565) | 0.366 | (-0.539 to 1.271) | 0.050 | (-0.273 to 0.373) |
| HR [1 bpm] | 0.292 | (0.070 to 0.513)* | -0.142 | (-0.345 to 0.061) | -0.156 | (-0.228 to -0.083)* |
| Hand grips [1 kg] | 0.060 | (-0.457 to 0.578) | 0.114 | (-0.360 to 0.588) | 0.174 | (0.005 to 0.343)* |
| Leg power [1 w] | -0.005 | (-0.015 to 0.006) | 0.001 | (-0.008 to 0.010) | 0.004 | (0.000 to 0.007)* |
| Trunk flexibility [1 cm] | -0.227 | (-0.493 to 0.039) | 0.062 | (-0.182 to 0.306) | 0.177 | (0.090 to 0.264)* |
| **Men** | ($n$ = 211 [1089 measurement]) | | | | | |
| Within R$^2$ | $R^2$ = 0.019 | | $R^2$ = 0.014 | | $R^2$ = 0.021 | |
| Age [1 year] | -0.234 | (-1.082 to 0.614) | 0.095 | (-0.672 to 0.862) | 0.064 | (-0.241 to 0.368) |
| Local area | -22.816 | (-46.72 to 1.088) | 28.225 | (6.332 to 50.118)* | -6.162 | (-14.207 to 1.884) |
| BMI [1 kg/m$^2$] | 4.178 | (1.109 to 7.248)* | -3.443 | (-6.203 to -0.684)* | -0.741 | (-1.890 to 0.409) |
| Waist/Hip ratio [1 point] | 27.048 | (-64.306 to 118.402) | -5.255 | (-86.431 to 75.921) | -23.299 | (-60.503 to 13.905) |
| Comorbidity score [1 point] | 1.049 | (-5.093 to 7.191) | 1.588 | (-3.881 to 7.056) | -2.848 | (-5.311 to -0.386)* |
| Smoker | -3.955 | (-20.252 to 12.342) | 9.425 | (-5.253 to 24.103) | -5.774 | (-11.788 to 0.241) |
| Alcohol intake [1% energy] | -0.408 | (-1.139 to 0.322) | 0.092 | (-0.560 to 0.744) | 0.299 | (0.011 to 0.587)* |
| Energy intake [1 kcal/day] | -0.009 | (-0.017 to -0.001)* | 0.006 | (-0.001 to 0.013) | 0.003 | (0.000 to 0.006)* |
| NRF 9.3 score [1 point] | 0.044 | (-0.021 to 0.109) | -0.029 | (-0.087 to 0.029) | -0.013 | (-0.039 to 0.013) |
| Hemoglobin [1 g/dl] | 1.209 | (-2.815 to 5.232) | -1.596 | (-5.174 to 1.982) | 0.356 | (-1.273 to 1.984) |
| HR [1 bpm] | 0.450 | (0.092 to 0.807)* | -0.284 | (-0.602 to 0.033) | -0.177 | (-0.323 to -0.031)* |
| Hand grips [1 kg] | -0.006 | (-0.807 to 0.795) | 0.350 | (-0.362 to 1.062) | -0.346 | (-0.672 to -0.021)* |
| Leg power [1 w] | -0.005 | (-0.02 to 0.011) | -0.001 | (-0.015 to 0.012) | 0.006 | (0.000 to 0.012)* |
| Trunk flexibility [1 cm] | 0.057 | (-0.418 to 0.531) | -0.101 | (-0.523 to 0.321) | 0.058 | (-0.134 to 0.250) |
| **Women** | ($n$ = 478 [2825 measurement]) | | | | | |
| Within R$^2$ | $R^2$ = 0.014 | | $R^2$ = 0.009 | | $R^2$ = 0.033 | |
| Age [1 year] | -0.703 | (-1.248 to -0.158)* | 1.166 | (0.666 to 1.665)* | -0.450 | (-0.609 to -0.292)* |
| Local area | -11.006 | (-29.129 to 7.117) | 22.441 | (5.983 to 38.899)* | -11.362 | (-16.637 to -6.087)* |
| BMI [1 kg/m$^2$] | 4.180 | (2.234 to 6.127)* | -3.410 | (-5.194 to -1.626)* | -0.768 | (-1.335 to -0.202)* |
| Waist/Hip ratio [1 point] | 45.384 | (-4.318 to 95.086) | -44.595 | (-90.603 to 1.413) | -0.196 | (-14.670 to 14.278) |
| Comorbidity score [1 point] | 4.938 | (1.089 to 8.787)* | -3.144 | (-6.704 to 0.417) | -1.809 | (-2.930 to -0.689)* |
| Smoker | 7.878 | (-2.663 to 18.418) | -6.799 | (-16.522 to 2.923) | -1.120 | (-4.190 to 1.949) |
| Alcohol intake [1% energy] | -0.404 | (-1.105 to 0.297) | 0.176 | (-0.471 to 0.824) | 0.218 | (0.014 to 0.422)* |
| Energy intake [1 kcal/day] | -0.002 | (-0.009 to 0.004) | 0.003 | (-0.003 to 0.009) | 0.000 | (-0.002 to 0.002) |
| NRF 9.3 score [1 point] | -0.016 | (-0.062 to 0.029) | -0.001 | (-0.043 to 0.041) | 0.018 | (0.004 to 0.031)* |
| Hemoglobin [1 g/dl] | -0.605 | (-1.776 to 0.565) | 0.645 | (-0.439 to 1.730) | -0.046 | (-0.387 to 0.295) |
| HR [1 bpm] | 0.184 | (-0.100 to 0.467) | -0.037 | (-0.299 to 0.225) | -0.146 | (-0.228 to -0.063)* |

*(Continued)*

**Table 2.** (Continued)

| Increment effects/unit | Inactive times | | LPA | | MVPA | |
|---|---|---|---|---|---|---|
| | RC | 95% CI | RC | 95% CI | RC | 95% CI |
| Hand grips [1 kg] | 0.126 | (-0.550 to 0.803) | -0.047 | (-0.673 to 0.580) | -0.076 | (-0.273 to 0.121) |
| Leg power [1 w] | -0.004 | (-0.019 to 0.012) | 0.000 | (-0.014 to 0.015) | 0.003 | (-0.001 to 0.008) |
| Trunk flexibility [1 cm] | -0.376 | (-0.703 to -0.049)* | 0.153 | (-0.149 to 0.456) | 0.223 | (0.127 to 0.318)* |

BMI, body mass index; CI, confidence interval; HR, heart rate; LPA, low intensity physical activity; MVPA, moderate-to-vigorous physical activity; NRF, nutrient-rich food; RC, regression coefficients. The results of these analyses are expressed as RC with 95% CI. The RC and 95% CI were calculated for changes in physical activity-related variables per unit increment for covariates.

Asterisk (*) indicates statistical significance ($p<0.05$). Sex and area were time-stable variables, and others were time-varying variables.

undertake activity education and assessment to prevent a rapid decrease in physical activity in older adults around the age of 70. Although cross-sectional studies have indicated that TEE and PAL measures using DLW methods are negatively associated with age around 60-years-old [37], these studies are limited to explaining population-level changes and cannot explain how an individual's TEE and PAL changes [38]. However, the triaxial accelerometer underestimated the measurements of TEE and PAL compared with those measured using the DLW technique, which is a gold standard in estimating TEE and PAL in participants with stable weight status [21]. Therefore, it is necessary to re-evaluate these results using TEE and PAL measured by the DLW method or with a more accurate accelerometer in a well-designed prospective repeated measurement study.

We indicated that there was no difference in correlation coefficients of physical activity-related variables between cross-sectional and repeated measurement; however, the cross-sectional analysis showed weaker correlations between TEE, inactive time, and LPA, as well as stronger correlations between TEE and chronological age compared with the repeated measurement analysis. Although TEE measured by DLW at baseline decreased significantly compared with that measured after 7 years in men, it did not change in women [39]. This is an important consideration since changes in fat-free mass were correlated with changes in resting metabolic rate for men but not for women [39]. In these relationships, it is possible that TEE does not substantially lower simply with increasing age because factors that change with age may be related to changes in TEE. Therefore, our results may suggest that TEE reflects the individual's LPA and inactive time changes, while the association between age and TEE was stronger while contributing to population-level changes than to individual-level changes.

Although many longitudinal studies have presented the association between predictors at baseline and physical activity [11–13], there is a need to evaluate the long-term longitudinal changes at the individual level in factors of dynamic changes associated with physical activity trajectory because an individual's habitual physical activity might be modified as time passes. Our results indicate that the MVPA trajectory was positively associated with alcohol consumption, hand grips, leg power, and trunk flexibility, while negatively associated with age, local area, BMI, comorbidity score, and HR over time. Although many previous studies have reported that higher MVPA is associated with lower body weight [40], HR [32], higher hand-grips, and the lower half of the body muscle power [33], these opposite relationships have not been well-elucidated [41]. It has been reported that maintaining or increasing time spent in MVPA is not beneficial to grip strength, whereas increasing grip strength increases spent time in MVPA [41]. It remains unclear whether the change in our indicated factors resulted in an increase in MVPA, because increasing MVPA may have resulted in changes in these factors.

**Table 3. Factors associated with changes of total energy expenditure, physical activity level, and step count by multivariate longitudinal analysis.**

| Increment effects/unit | TEE | | PAL [a] | | Step counts | |
|---|---|---|---|---|---|---|
| | RC | 95% CI | RC | 95% CI | RC | 95% CI |
| **Total** | ($n$ = 689 [3914 measurement]) | | | | | |
| Within R$^2$ | $R^2$ = 0.016 | | $R^2$ = 0.015 | | $R^2$ = 0.079 | |
| Age [1 year] | -5.677 | (-6.772 to -4.582)* | -0.012 | (-0.081 to 0.058) | -89.689 | (-106.981 to -72.293)* |
| Women sex | -289.060 | (-323.048 to -255.072)* | 7.280 | (5.011 to 9.547)* | -624.138 | (-1182.056 to -66.219)* |
| Local area | 2.336 | (-29.295 to 33.968) | 0.766 | (-1.366 to 2.898) | -1847.742 | (-2371.960 to -1323.523)* |
| BMI [1 kg/m$^2$] | 20.190 | (16.176 to 24.202)* | -0.557 | (-0.810 to -0.302)* | -112.385 | (-175.195 to -49.574)* |
| Waist/Hip ratio [1 point] | -41.811 | (-161.007 to 77.385) | -8.154 | (-15.126 to -1.182)* | -2321.310 | (-4059.591 to -583.028)* |
| Comorbidity score [1 point] | -10.486 | (-19.303 to -1.667)* | -0.719 | (-1.238 to -0.198)* | -246.592 | (-376.104 to -117.078)* |
| Smoker | -9.186 | (-31.753 to 13.380) | -1.131 | (-2.515 to 0.254) | -231.790 | (-575.509 to 111.928) |
| Alcohol intake [1% energy] | 2.176 | (0.831 to 3.521)* | 0.096 | (0.014 to 0.176)* | 41.491 | (21.441 to 61.541)* |
| Energy intake [1 kcal/day] | 0.017 | (0.002 to 0.030)* | 0.001 | (0.000 to 0.001)* | 0.096 | (-0.108 to 0.300) |
| NRF 9.3 score [1 point] | 0.060 | (-0.042 to 0.161) | 0.004 | (-0.002 to 0.009) | 1.081 | (-0.412 to 2.573) |
| Hemoglobin [1 g/dl] | 1.523 | (-1.213 to 4.259) | 0.083 | (-0.076 to 0.241) | -14.066 | (-53.766 to 25.634) |
| HR [1 bpm] | -1.104 | (-1.713 to -0.493)* | -0.069 | (-0.104 to -0.033)* | -12.889 | (-21.774 to -4.002)* |
| Hand grips [1 kg] | 1.625 | (0.203 to 3.046)* | -0.056 | (-0.139 to 0.027) | 34.294 | (6.441 to 62.882)* |
| Leg power [1 w] | 0.027 | (0.009 to 0.046)* | 0.001 | (-0.001 to 0.002) | 0.128 | (-0.297 to 0.554) |
| Trunk flexibility [1 cm] | 1.424 | (0.694 to 2.153)* | 0.042 | (-0.001 to 0.085) | 14.008 | (3.334 to 24.681)* |
| **Men** | ($n$ = 211 [1089 measurement]) | | | | | |
| Within R$^2$ | $R^2$ = 0.002 | | $R^2$ = 0.009 | | $R^2$ = 0.047 | |
| Age [1 year] | -8.313 | (-10.648 to -5.978)* | -0.013 | (-0.154 to 0.128) | -33.674 | (-67.961 to 0.613) |
| Local area | 31.178 | (-28.730 to 91.085) | 2.718 | (-1.003 to 6.439) | -1250.891 | (-2152.825 to -348.956)* |
| BMI [1 kg/m$^2$] | 16.847 | (7.815 to 25.879)* | -0.621 | (-1.151 to -0.090)* | -153.350 | (-283.270 to -23.430)* |
| Waist/Hip ratio [1 point] | 106.633 | (-203.035 to 416.301) | -11.166 | (-28.285 to 5.953) | -5440.193 | (-9678.728 to -1201.657)* |
| Comorbidity score [1 point] | -7.602 | (-27.839 to 12.634) | -0.525 | (-1.659 to 0.609) | -376.418 | (-656.465 to -96.370)* |
| Smoker | 3.392 | (-43.322 to 50.105) | -0.308 | (-3.084 to 2.468) | -427.694 | (-1106.471 to 251.083) |
| Alcohol intake [1% energy] | 1.835 | (-0.495 to 4.166) | 0.121 | (-0.011 to 0.254) | 37.918 | (5.268 to 70.569)* |
| Energy intake [1 kcal/day] | 0.038 | (0.012 to 0.065)* | 0.002 | (0.001 to 0.004)* | 0.319 | (-0.048 to 0.687) |
| NRF 9.3 score [1 point] | -0.170 | (-0.386 to 0.046) | -0.005 | (-0.017 to 0.007) | -1.532 | (-4.516 to 1.452) |
| Hemoglobin [1 g/dl] | -4.713 | (-18.2 to 8.775) | -0.294 | (-1.044 to 0.455) | 19.407 | (-166.018 to 204.832) |
| HR [1 bpm] | -1.050 | (-2.267 to 0.167) | -0.114 | (-0.182 to -0.047)* | -18.855 | (-35.487 to -2.223)* |
| Hand grips [1 kg] | -0.815 | (-3.518 to 1.888) | -0.115 | (-0.264 to 0.035) | 0.653 | (-36.400 to 37.706) |
| Leg power [1 w] | 0.015 | (-0.036 to 0.067) | 0.002 | (-0.001 to 0.005) | 0.538 | (-0.171 to 1.247) |
| Trunk flexibility [1 cm] | 0.496 | (-1.092 to 2.085) | -0.017 | (-0.105 to 0.072) | -0.545 | (-22.393 to 21.302) |
| **Women** | ($n$ = 478 [2825 measurement]) | | | | | |
| Within R$^2$ | $R^2$ = 0.054 | | $R^2$ = 0.017 | | $R^2$ = 0.079 | |
| Age [1 year] | -5.378 | (-6.413 to -4.342)* | -0.033 | (-0.114 to 0.048) | -90.658 | (-107.948 to -73.367)* |
| Local area | -20.452 | (-54.194 to 13.290) | -0.598 | (-3.265 to 2.068) | -1763.020 | (-2282.308 to -1243.732)* |
| BMI [1 kg/m$^2$] | 20.250 | (16.550 to 23.949)* | -0.497 | (-0.786 to -0.208)* | -105.660 | (-168.230 to -43.090)* |
| Waist/Hip ratio [1 point] | -13.582 | (-110.199 to 83.036) | -6.971 | (-14.419 to 0.477) | -2350.588 | (-4089.658 to -611.518)* |
| Comorbidity score [1 point] | -11.852 | (-19.324 to -4.380)* | -0.809 | (-1.385 to -0.233)* | -238.126 | (-367.479 to -108.773)* |
| Smoker | -8.592 | (-28.915 to 11.732) | -1.584 | (-3.159 to -0.010)* | -128.181 | (-459.371 to 203.009) |
| Alcohol intake [1% energy] | 1.511 | (0.154 to 2.868)* | 0.075 | (-0.030 to 0.179) | 43.815 | (23.864 to 63.766)* |
| Energy intake [1 kcal/day] | 0.002 | (-0.010 to 0.015) | 0.000 | (-0.001 to 0.001) | 0.127 | (-0.075 to 0.330) |
| NRF 9.3 score [1 point] | 0.097 | (0.009 to 0.186)* | 0.007 | (0.001 to 0.014)* | 1.007 | (-0.485 to 2.499) |
| Hemoglobin [1 g/dl] | 0.197 | (-2.082 to 2.475) | 0.070 | (-0.106 to 0.246) | -15.921 | (-55.666 to 23.823) |
| HR [1 bpm] | -0.473 | (-1.024 to 0.078) | -0.041 | (-0.083 to 0.002) | -13.007 | (-21.896 to -4.119)* |

(*Continued*)

**Table 3.** (Continued)

| Increment effects/unit | TEE | | PAL [a] | | Step counts | |
|---|---|---|---|---|---|---|
| | RC | 95% CI | RC | 95% CI | RC | 95% CI |
| Hand grips [1 kg] | -0.568 | (-1.883 to 0.748) | -0.026 | (-0.127 to 0.076) | 16.928 | (-3.693 to 37.550) |
| Leg power [1 w] | 0.029 | (0.000 to 0.059)* | 0.001 | (-0.001 to 0.003) | 0.164 | (-0.261 to 0.589) |
| Trunk flexibility [1 cm] | 1.119 | (0.484 to 1.754)* | 0.070 | (0.021 to 0.119)* | 12.541 | (1.943 to 23.139)* |

BMI, body mass index; CI, confidence interval; HR, heart rate; NRF, nutrient-rich food; PAL; physical activity level; RC, regression coefficients; TEE, total energy expenditure. The results of these analyses are expressed as RC with 95% CI. The RC and 95% CI were calculated for changes in physical activity-related variables per unit increment for covariates.

Asterisk (*) indicates statistical significance ($p < 0.05$). Sex and area were time-stable variables, and others were time-varying variables.

[a] The RC and 95% CI shown in the estimated value corrected by the $10^2$ because the estimated value was small.

However, regardless of which comes first, a better physical activity trajectory [9] and improving these associated factors [42–44] are both important for preventing risks of all-cause and cause-specific mortality. Previous cross-sectional studies have reported that body flexibility is not associated with moderate physical activity in older adults [34]. However, we demonstrated that higher trunk flexibility was related to higher MVPA in this longitudinal study. A previous cross-sectional study has reported that weekly alcohol consumption was associated with a higher level of physical activity among young, middle-aged, and older adults [45]. The detailed mechanisms and causal relationships must be clarified with further intervention and basic studies. In addition, it has been shown that increasing age [8], local area [11], and comorbidity [13] are associated with lower physical activity, which supports our results. A longitudinal study of healthy older participants has reported that maintaining or increasing activity levels prevents or attenuates physiological function decline with age [46]. Therefore, it may potentially be useful to advise on healthy behavior and factor changes based on our findings to help increase and maintain physical activity or physiological function in adults.

The strength of the present study is the objective measurement of repetitive physical activity and covariates using a validated tool in the same population. This approach provides an opportunity to determine the association of changes in different parameters of dynamic changes associated with physical activity trajectory across a wide age range. We were able to minimize the influence of confounders derived from interindividual variance by using the repeated data described above. However, the present study has certain methodological limitations. First, the participants may be more health-conscious than the general Japanese population because we could not include study participants by a random sampling method. In fact, several thousands more daily step counts were observed in the present study than in the National Health and Nutrition Examination Survey in Japan examined by random sampling [47]. In addition, the participant's characteristics, such as age and physical activity including step count and LPA, differed from those of individuals who were excluded from the present study. Therefore, the present study may have a selection bias. Second, this study had a relatively short follow-up period, and this might have influenced the physical activity trajectory. However, the sample size and frequency of surveys required for our group's mean physical activity trajectory were adequate. Third, we were unable to completely eliminate confounding factors and measurement bias. Although we examined economic and medication status, we could not use these variables as covariates due to incomplete data. These limitations may affect the generalization of our results. Therefore, a well-designed longitudinal cohort study with larger randomized samples is required in the future to further determine the factors of dynamic changes associated with changes in the objectively measured physical activity.

## Conclusions

Our results indicate that MVPA, step count, PAL, and TEE trajectory show clear curvature and accelerated rate of change around the age of 70. Moreover, these determined physical health, fitness, and BMI as factors of dynamic changes associated with physical activity changes. Given the insufficient physical activity and heterogeneity of physical activity across regions and countries, there is an urgent need to highlight the importance of improving factors associated with greater beneficial effects for increasing physical activity worldwide. Therefore, these results may provide useful insights into the development of effective strategies for increasing physical activity.

## Supporting information

**S1 File.**
(DOCX)

## Acknowledgments

We are grateful to all of the participants who provided data for use in this research and to the members of the Physical Activity Research Laboratory at the National Institute of Health and Nutrition. The authors thank Dr. Michiya Tanimoto, Dr. Noriko Tanaka, Dr. Hiroshi Kawano, Dr. Kenta Yamamoto, Dr. Motoyuki Iemitsu, Dr. Kiyoshi Sanada, Ms. Yumi Ohmori, Ms. Rie Katayama, Dr. Zhenbo Cao, Ms. Eriko Kubo, Ms. Miyuki Hayashi, Mr. Satoshi Hanawa, Ms. Naeko Kurose, Ms. Aiko Hirosako, Ms. Sayaka Nakamura, Ms. Hidemi Hara, Ms. Miki Yoshida, Dr. Satoshi Kurita, Ms. Noriko Wada, Ms. Miho Okamoto, Ms. Hisako Ito, Ms. Kinue Nakajima, Ms. Kaori Sato, and Ms. Kazumi Kajiwara, who significantly contributed to the realization of this cohort study through their long-term involvement as research assistants.

## Author Contributions

**Conceptualization:** Daiki Watanabe, Motohiko Miyachi.

**Data curation:** Haruka Murakami, Yuko Gando, Ryoko Kawakami, Kumpei Tanisawa, Harumi Ohno, Kana Konishi, Azusa Sasaki, Akie Morishita, Motohiko Miyachi.

**Formal analysis:** Daiki Watanabe.

**Funding acquisition:** Motohiko Miyachi.

**Project administration:** Haruka Murakami, Akie Morishita.

**Supervision:** Nobuyuki Miyatake, Motohiko Miyachi.

**Visualization:** Daiki Watanabe, Motohiko Miyachi.

**Writing – original draft:** Daiki Watanabe, Yuko Gando, Motohiko Miyachi.

**Writing – review & editing:** Haruka Murakami, Ryoko Kawakami, Kumpei Tanisawa, Harumi Ohno, Kana Konishi, Azusa Sasaki, Nobuyuki Miyatake.

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
