## [Decision Letter · Decision Letter 0]

23 Nov 2022

PONE-D-22-03855Factors associated with changes in the objectively measured physical activity among Japanese adults: A longitudinal and dynamic panel data analysisPLOS ONE

Dear Dr. Miyachi,

Thank you for submitting your manuscript to PLOS ONE. After careful consideration, we feel that it has merit but does not fully meet PLOS ONE’s publication criteria as it currently stands. Therefore, we invite you to submit a revised version of the manuscript that addresses the points raised during the review process.

We look forward to receiving your revised manuscript.

Kind regards,

Hsin-Yen Yen

Academic Editor

PLOS ONE

Reviewers' comments:

Reviewer's Responses to Questions

**Comments to the Author**

1. Is the manuscript technically sound, and do the data support the conclusions?

Reviewer #1: Yes

Reviewer #2: Yes

2. Has the statistical analysis been performed appropriately and rigorously? 

Reviewer #1: Yes

Reviewer #2: Yes

3. Have the authors made all data underlying the findings in their manuscript fully available?

Reviewer #1: Yes

Reviewer #2: Yes

4. Is the manuscript presented in an intelligible fashion and written in standard English?

Reviewer #1: Yes

Reviewer #2: Yes

5. Review Comments to the Author

Reviewer #1: This longitudinal study looks at changes in objectively measured physical activity (LPA, MVPA, TEE, PAL, steps per day) over time among Japanese adults, as well as factors that are associated with those changes. The exploration of changes in objective measures of physical activity within the same individuals over time is a key strength of this study, as most studies rely on self-reported measures of physical activity and/or cross-sectional data collection.

Major comments

• Given the gender differences in physical activity levels, trajectories (particularly for LPA), and distributions of predictor variables/covariates that were included in the models, it surprised me that the models exploring the factors associated with changes in physical activity were not conducted separately for men and women. Is it possible to stratify the models by gender?

• ‘Inactive time’ here is the combination of nonwear time (including sleep) and sedentary time. This is problematic as these are quite different constructs, and it makes it difficult to discern whether changes in inactive time reflect changes in sedentary time, sleep, nonwear, or all three. If possible, sedentary time should be modeled as an outcome on its own.

• I’m not sure I understand exactly what insights the different groups in the latent class growth models are meant to provide, and I particularly do not understand how the interpretation of the lack of impact of the counseling on MVPA (lines 238-241) relates to FigureS2C. Could the authors please elaborate?

Minor comments

• Could you please clarify what the minimum wear criteria were for the accelerometer data? Lines 129-131 imply that at least 7 days of complete wear (≥6 hours) were required but it isn’t quite clear. Additionally, did any participants who wore the accelerometer fail to meet the minimum wear criteria?

• Please add a comment regarding how many valid days of accelerometer data participants provided at each time point. The range between how many days participants were asked to wear it (28) and the minimum wear criteria (presumably 7) is quite large so it would be useful to know how many days of wear on average the data are drawn from.

• Please clarify the unit of measurement of alcohol intake (1% energy) and how it was derived to help with the interpretation of the regression coefficient.

• The positive association between alcohol intake and MVPA is surprising and interesting but is not addressed in the discussion. Could the authors please comment on this in the discussion?

• Some of the predictor variables in the multivariate longitudinal analysis seem possibly strongly correlated with each other (e.g., BMI and waist/hip ratio). Were the models checked for collinearity?

Reviewer #2: The findings from the study are very useful for everyone to project what changes occur when an individual ages especially in physical activity. The paper is well-written, and the reading is easy to follow. The emphasis on objectively measured physical activity in Japan measurement and its correlation with other factors that take change during the ageing process. The authors have explained the factors involved and their influence on it statistically. To my understanding, this is an (interesting) seminal work, that can produce interesting results if can be compared with other countries too.

6. PLOS authors have the option to publish the peer review history of their article (what does this mean?). If published, this will include your full peer review and any attached files.

Reviewer #1: No

Reviewer #2: **Yes: **Mahenderan Appukutty

---

## [Author Response · Author response to Decision Letter 0]

7 Dec 2022

Responses to the comments of Reviewer 1

We thank the Reviewer for the constructive comments regarding our paper. We have incorporated changes that reflect the suggestions the editor has graciously provided. We hope that our changes and the responses we provide below satisfactorily address all the issues and concerns that have been raised. 

Major comments

Comment #1 (Results): Given the gender differences in physical activity levels, trajectories (particularly for LPA), and distributions of predictor variables/covariates that were included in the models, it surprised me that the models exploring the factors associated with changes in physical activity were not conducted separately for men and women. Is it possible to stratify the models by gender?

Our response #1: Thank you very much for this comment. We agreed with your comments. We have added this information in the Table 2 and 3 and Supplementary Table 5 and Results in accordance to comment, as follows: “These associations such as inactive times, LPA, TEE, PAL, and step counts with covariates were confirmed sex differences.” (page 19, lines 299-300). 

Comment #2 (Results): Inactive time’ here is the combination of nonwear time (including sleep) and sedentary time. This is problematic as these are quite different constructs, and it makes it difficult to discern whether changes in inactive time reflect changes in sedentary time, sleep, nonwear, or all three. If possible, sedentary time should be modeled as an outcome on its own.

Our response #2: We thank the Reviewer for this comment. We agreed with your comments. We have added this information in the Supplementary Table 5 and Results in accordance to comment, as follows: “Both inactive and sedentary times trajectory are associated with higher BMI (Table 2 and Table S5).” (page 19, lines 294-295).

Comment #3 (Results): I’m not sure I understand exactly what insights the different groups in the latent class growth models are meant to provide, and I particularly do not understand how the interpretation of the lack of impact of the counseling on MVPA (lines 238-241) relates to FigureS2C. Could the authors please elaborate?

Our response #3: Thank you very much for this comment. We have modified this information in the Results in accordance to comment, as follows: “The MVPA trajectory were classified into two groups (Figure S1C and S2C; 1: low MVPA trajectory group and 2: high MVPA trajectory group). Although this cohort study had a brief counseling intervention for physical activity in some populations, the proportion of individuals in the high MVPA trajectory group in the intervention group was similar to that in the non-intervention group (5.1% vs 4.9%). These results suggest that brief counseling intervention did not affect the MVPA trajectory.” (page 15-16, lines 258-263).

Minor comments

Comment #4 (Methods): Could you please clarify what the minimum wear criteria were for the accelerometer data? Lines 129-131 imply that at least 7 days of complete wear (≥6 hours) were required but it isn’t quite clear. Additionally, did any participants who wore the accelerometer fail to meet the minimum wear criteria?

Our response #4: We thank the Reviewer for this comment. We agreed with your comments. We have added this information in the Methods section in accordance to comment, as follows: “The participants were asked to wear a triaxial accelerometer at least more than 10 hours per day for 28 days. The days were considered valid when participants wore the wearable devices for more than 10 hours/day from self-reported wearing time based on activity records by the participant himself/herself [22]. However, the objective wear time obtained by accelerometer does not include acceleration data with an intensity of less than 1.1 METs when a participant is completely still (no signal time). Therefore, the calculated wearing time cannot distinguish between no signal time and non-wearing time. The objective wearing time was defined as 24 hours minus non-wearing and no signal time. In addition, we excluded data with wearing time less than 6 hours per day. Those who exceeded this criterion confirmed self-reported accelerometer wearing time exceeding 10 hours in 24 individuals randomly selected from the study population. Therefore, we ultimately determined a valid days of accelerometer data use from two criteria: self-reported wearing time from activity records, and objective wearing time from the accelerometer. Median valid days of accelerometer data included more than 2 weeks in all in-person testing from 2007 to 2018 (Table S1). To calculate the mean physical activity time, the sum of all physical activities surveyed over at last 7 days (including weekdays and weekends) was divided by the number of adhered days. No participants were excluded by these criteria for accelerometer data because those who had the valid days of accelerometer data for less than 7 days were asked to wear the accelerometer again.” (pages 8-9, lines 125-143).

Reference

22. Tudor-Locke C, Camhi SM, Troiano RP. A catalog of rules, variables, and definitions applied to accelerometer data in the National Health and Nutrition Examination Survey, 2003-2006. Prev Chronic Dis. 2012;9:E113. doi: 10.5888/pcd9.110332. PubMed PMID: 22698174; PubMed Central PMCID: PMCPMC3457743.

Comment #5 (Methods, Page 5): Please add a comment regarding how many valid days of accelerometer data participants provided at each time point. The range between how many days participants were asked to wear it (28) and the minimum wear criteria (presumably 7) is quite large so it would be useful to know how many days of wear on average the data are drawn from.

Our response #5: Thank you very much for this comment. We have modified this information in the Supplementary Table 1 and Methods in accordance to comment, as follows: “Median valid days of accelerometer data included more than 2 weeks in all in-person testing from 2007 to 2018 (Table S1). To calculate the mean physical activity time, the sum of all physical activities surveyed over at last 7 days (including weekdays and weekends) was divided by the number of adhered days. No participants were excluded by these criteria for accelerometer data because those who had the valid days of accelerometer data for less than 7 days were asked to wear the accelerometer again.” (page 9, lines 138-143).

Comment #6 (Methods, Page 6, lines 86): Please clarify the unit of measurement of alcohol intake (1% energy) and how it was derived to help with the interpretation of the regression coefficient.

Our response #6: We thank the Reviewer for this comment. We have added this information in the Methods section in accordance to comment, as follows: “Alcohol intake was estimated from the consumption frequency and portion size of each alcohol beverage using a program developed based on the Standard Tables of Food Composition in Japan. Alcohol intake (%) was calculated by dividing energy intake from alcohol by total energy intake and multiplying by 100.” (page 10, lines 161-164).

Comment #7 (Methods, Page 7): The positive association between alcohol intake and MVPA is surprising and interesting but is not addressed in the discussion. Could the authors please comment on this in the discussion?

Our response #7: Thank you very much for this comment. We agreed with your comments. We have added this information in the Discussion section in accordance to comment, as follows: “A previous cross-sectional study has reported that weekly alcohol consumption was associated with a higher level of physical activity among young, middle-aged, and older adults [45]. The detailed mechanisms and causal relationships must be clarified with further intervention and basic studies.” (pages 22-23, lines 365-368)

Reference

45. Werneck AO, Oyeyemi AL, Szwarcwald CL, Silva DR. Association between physical activity and alcohol consumption: sociodemographic and behavioral patterns in Brazilian adults. J Public Health (Oxf). 2019;41(4):781-7. doi: 10.1093/pubmed/fdy202. PubMed PMID: 30445471.

Comment #8 (Methods, Page 7): Some of the predictor variables in the multivariate longitudinal analysis seem possibly strongly correlated with each other (e.g., BMI and waist/hip ratio). Were the models checked for collinearity?

Our response #8: We thank the Reviewer for this comment. We agreed with your comments. We have added this information in the Supplementary Table 2 and Methods section in accordance to comment, as follows: “The variance inflation factor (VIF) was used to avoid multicollinearity in the multivariate regression model and all covariates were maintained VIF ≤5 (Table S2).” (page 13, lines 220-222) 

Responses to the comments of Reviewer 2

We thank the Reviewer for the constructive comments regarding our paper. We have incorporated changes that reflect the suggestions the editor has graciously provided. We hope that our changes and the responses we provide below satisfactorily address all the issues and concerns that have been raised. 

Comment #1 (Overall): The findings from the study are very useful for everyone to project what changes occur when an individual ages especially in physical activity. The paper is well-written, and the reading is easy to follow. The emphasis on objectively measured physical activity in Japan measurement and its correlation with other factors that take change during the ageing process. The authors have explained the factors involved and their influence on it statistically. To my understanding, this is an (interesting) seminal work, that can produce interesting results if can be compared with other countries too.

Our response #1: We thank the Reviewer for this comment. We agreed with your comments. Unfortunately, any study similar to our study on trajectory analysis of physical activity using accelerometer has not yet been conducted in other countries. It is therefore interesting to compare our Japanese findings with other countries, but at present it is not possible. Again, thank you for reviewing for our manuscript.

---

## [Editor Report · Decision Letter 1]

12 Jan 2023

Factors associated with changes in the objectively measured physical activity among Japanese adults: A longitudinal and dynamic panel data analysis

PONE-D-22-03855R1

Dear Dr. Miyachi,

We’re pleased to inform you that your manuscript has been judged scientifically suitable for publication and will be formally accepted for publication once it meets all outstanding technical requirements.

Kind regards,

Hsin-Yen Yen

Academic Editor

PLOS ONE
---

## [Editor Report · Acceptance letter]

7 Feb 2023

PONE-D-22-03855R1 

Factors associated with changes in the objectively measured physical activity among Japanese adults: A longitudinal and dynamic panel data analysis 

Dear Dr. Miyachi:

I'm pleased to inform you that your manuscript has been deemed suitable for publication in PLOS ONE. Congratulations! Your manuscript is now with our production department. 

Kind regards, 

on behalf of

Dr. Hsin-Yen Yen 

Academic Editor

PLOS ONE